# Alpha2 Adrenergic Modulation of Spike-Wave Epilepsy: Experimental Study of Pro-Epileptic and Sedative Effects of Dexmedetomidine

**DOI:** 10.3390/ijms24119445

**Published:** 2023-05-29

**Authors:** Evgenia Sitnikova, Maria Pupikina, Elizaveta Rutskova

**Affiliations:** Institute of the Higher Nervous Activity and Neurophysiology of Russian Academy of Sciences, Butlerova Str., 5A, 117485 Moscow, Russiaerutskova@gmail.com (E.R.)

**Keywords:** absence epilepsy, spike-wave discharges, rat model, in vivo, alpha2 adrenergic receptors, dexmedetomidine, automatic detection, cluster analysis

## Abstract

In the present report, we evaluated adrenergic mechanisms of generalized spike-wave epileptic discharges (SWDs), which are the encephalographic hallmarks of idiopathic generalized epilepsies. SWDs link to a hyper-synchronization in the thalamocortical neuronal activity. We unclosed some alpha2-adrenergic mechanisms of sedation and provocation of SWDs in rats with spontaneous spike-wave epilepsy (WAG/Rij and Wistar) and in control non-epileptic rats (NEW) of both sexes. Dexmedetomidine (Dex) was a highly selective alpha-2 agonist (0.003–0.049 mg/kg, i.p.). Injections of Dex did not elicit de novo SWDs in non-epileptic rats. Dex can be used to disclose the latent form of spike-wave epilepsy. Subjects with long-lasting SWDs at baseline were at high risk of absence status after activation of alpha2- adrenergic receptors. We create the concept of alpha1- and alpha2-ARs regulation of SWDs via modulation of thalamocortical network activity. Dex induced the specific abnormal state favorable for SWDs—“alpha2 wakefulness”. Dex is regularly used in clinical practice. EEG examination in patients using low doses of Dex might help to diagnose the latent forms of absence epilepsy (or pathology of cortico-thalamo-cortical circuitry).

## 1. Introduction

It is widely accepted that the brain’s noradrenergic system regulates working memory, attention function and arousal level [1,2,3]. It also protects against epileptic seizures [4]. However, the opposite pro-epileptic properties of noradrenaline (NA), as reviewed in [5], suggest that the role of NA in epilepsy is not obvious.

The center of the brain’s noradrenergic system lies in the Locus Coeruleus (LC), which sends wide connections throughout the brain [1]. It controls arousal and attention and modulates the neuronal activity of the cortico-thalamo-cortical circuitry [3,6,7,8]. The cortico-thalamo-cortical circuitry is involved in information processing during wakefulness and generates spontaneous rhythmic activity during sleep, such as sleep spindles and delta waves [8,9,10,11]. It is known that the cortico-thalamo-cortical neuronal network could produce generalized spike-wave epileptic discharges (SWDs), which are the encephalographic hallmarks of idiopathic generalized epilepsies [12,13,14,15]. Mechanisms of SWDs have been thoroughly studied: SWDs are caused by inherited channelopathies and neuronal and network pathologies [10,16,17,18,19,20]. SWDs are also known to be effectively modulated by noradrenergic mechanisms [21,22,23,24,25,26].

There are three main types of adrenergic receptors (ARs): alpha1, alpha2 and beta [3,27,28]. These ARs types have different affinities to NA and are non-homogeneously distributed over the brain [28,29,30,31]. Alpha2-ARs are known to have the highest affinity for NA (~50 nM) compared to alpha1 and beta ARs [21]. On the other hand, beta-ARs are the least sensitive to NA and do not seem to play a significant role in generating epileptic SWDs [21,32]. Buzsaki et al., in their study [21], discussed the role of alpha1- and alpha2-ARs in pathological epileptic brain activity generated in the cortico-thalamo-cortical network. Based on their findings, they considered the thalamus a neural network where anti-epileptic alpha1-ARs activation and pro-epileptic alpha2-ARs activation could balance each other.

Dexmedetomidine (Dex) is a highly selective alpha2-ARs agonist used in clinical and veterinary practice as a sedative pharmacological agent [33,34,35,36]. In their review paper in 2021, Lui et al. [37] characterized Dexmedetomidine as “*sedative, anxiolytic, analgesic, sympatholytic, and opioid-sparing properties and induces a unique sedative response which shows an easy transition from sleep to wakefulness*”. Less attention is given to Dex (like other alpha2-ARs agonists) promoting SWDs. Rats with spontaneous SWDs showed an elevation of spike-wave activity after the administration of Dex (Table 1). 

Administration of Dex in a very high dose led to a decrease in the number of SWDs. Biphasic enhancement of spike-wave epileptic activity in WAG/Rij rats after Dex injection has recently been reported [22,23]. A biphasic character of SWDs could be accounted for by the sedative effect of Dex, as long as the sedative state has been induced soon after the initial enhancement of SWDs. The analyzed period after Dex injection might include (at least partly) the sedation period when SWDs are blocked. Therefore, an increase or a decrease in SWDs could be detected depending on sedation depth and duration.

In WAG/Rij rats, SWDs mainly occur during the transition from wakefulness to sleep [39,40]. During this transition, neuronal firing in the LC gradually decreases, and NA release is slightly reduced. This leads to lower activation of ARs in target cells [1,3,41]. Considering that alpha2-ARs are more sensitive to NA than alpha1- and beta-ARs [32], we hypothesize that the alpha2-mediated effect of NA would be greater than that mediated by other types of ARs. In addition to the decreased counteraction from thalamic alpha1-ARs, this temporally promotes the generation of SWDs in the cortico-thalamo-cortical circuit.

Further influence of brain sleep-promoting systems leads to the predominance of slow-wave (delta) oscillations in the cortico-thalamo-cortical network. Figure 1 illustrates that the cortico-thalamo-cortical network readily generates SWDs when a certain ratio of alpha2/alpha1-ARs activation is achieved. During wakefulness, alpha1- and alpha2-AR activities are in a state of dynamic equilibrium. The transition from wakefulness to sleep temporarily brings it to a value that is optimal for the occurrence of SWDs. Similar changes in the activation of alpha-ARs should occur after administering alpha2-AR agonists leading to the generation of SWDs (the bottom insertion in Figure 1). Agonists of alpha2-ARs might directly interact with the thalamic ARs. At the same time, activation of thalamic alpha1-ARs should be reduced via an indirect pathway. This would be accomplished by activating presynaptic alpha2-ARs in the LC, reducing its neuronal firing and subsequent release of NA to target cells’ receptors [42,43].

The agonist of alpha2-ARs, clonidine, systemically injected in the WAG/Rij rat model of absence epilepsy in a very low dose (0.00625 mg/kg) caused prolonged and recurrent spike-wave seizures [24]. The agonist of alpha2-ARs, Dex, has recently been shown to trigger absence status in genetic absence epilepsy rats from Strasbourg (GAERS) [44]. The tremendous increase in SWDs elicited with agonists of alpha2-ARs could meet the criteria for absence status epilepticus (AS): “*a pro-longed, generalized, and nonconvulsive seizure recurrent, unprovoked episodes of typical absence seizures”* [45]. Pharmacologically induced status epilepticus by activating alpha2-ARs relates to *“acute symptomatic epilepsy as a reaction to the drug*” [46]. Here we aimed to describe the temporal dynamics of Dex epileptic activity provocation in WAG/Rij rats in consideration of sedative state induction. We hypothesized that low doses of Dex would lead to prolonged and recurrent spike-wave epileptic activity by turning the alpha2/alpha1-ARs index to the values favorable for SWDs but being too low to induce a sedative state. Another assumption to be tested in the current study was that a pro-absence effect of Dex could be found only in epileptic subjects. In healthy non-epileptic subjects’ injections of Dex cannot induce de novo SWDs. This experimental study was performed in adult rats aged six months and older when genuine SWDs were known to appear spontaneously [47,48,49,50].

## 2. Results

We performed in vivo pharmaco-EEG investigation in 26 adult rats. All of the rats were bred and raised at our Institute, and they were all intact and not genetically modified. We had 11 WAG/Rij rats (nine females and two males) and 1 Wistar rat (female). The WAG/Rij with spontaneous SWDs and Wistar rats are the most commonly used rat strains in research [49,51,52,53]. The NEW substrain has been selected since 2016 as a non-epileptic control for WAG/Rij rats [54]. In the NEW substrain, spontaneous SWDs were absent during life [55]. Electrical brain activity was recorded in freely moving rats via implanted epidural EEG electrodes. The EEG was recorded for 6–24 h starting in the middle of the light phase (from 1 to 4 p.m.). The genuine spike-wave discharges (SWDs) were automatically detected in the full-length EEG records using a wavelet-based approach [56] (see details in Appendix A). The inclusion criterion for the epileptic group: higher than three genuine SWDs per hour in a full-length baseline record. There were no exclusion criteria for the epileptic group. This group included all 11 WAG/Rij rats, 1 NEW and 1 Wistar (13 rats in total, Figure 2a, and Appendix A). 

The control group consisted of age-matched rats. The inclusion criterion for the non-epileptic group: the number of genuine SWDs in baseline was less than 0.25 per hour (1 SWD per 4 h). The exclusion criterion: the number of genuine SWDs in baseline was higher than 3 per hour. In total, 13 subjects of the NEW substrain were included; subject M5 of the NEW substrain met the exclusion criterion and was moved to the epileptic group (Appendix A).

To quickly and safely reduce the number of rats used for testing, rats were used for multiple tests with 14–21 days intervals between tests. Therefore, 1–3 baseline EEGs and 1–3 pharmaco-EEG tests were performed in each rat. A baseline recording preceded each pharmaco-EEG recording. Dexmedetomidine (Dex) was i.p. injected in doses 0.003–0.049 mg/kg (from 0.01 to 0.12 mL of Dexdomitor^®^, Orion Pharma, Espoo, Finland). The estral phase was examined in female subjects. The phases of the estral cycle did not affect the parameters of SWDs.

SWDs were detected when wavelet power at 8–10 Hz and 17–20 Hz exceeded individually chosen threshold values (Appendix A). The threshold values selected to detect SWDs in baseline were also used to detect SWDs in Dex conditions (Section 4.3.2). This provided accurate and reliable detections, which helped to overcome the significant drawback of subjectivity. We analyzed datasets containing the start and end points of SWDs and computed the following scores: The density of SWDs (dSWD) was computed as the duration of SWDs in seconds per minute for 5 min and 1 h bins.The number of SWDs (nSWD) per 1 h.The mean duration of SWDs (durSWD) per 1 h.

To further analyse Dex effects in the epileptic group for each animal, we computed the difference of dSWD between baseline and Dex conditions (differential dSWD).

### 2.1. The Effect of Dexmedetomidine in Non-Epileptic Control Rat Subjects

This part of the study was conducted in 13 non-epileptic control rats of both sexes (8 females and five males, NEW substrain, Appendix A). Spontaneous SWDs in these rats were extremely rare (the median of nSWD was 0.049/per hour with Q1 = 0 and Q3 = 0.092; Appendix A). 

I.p. injections of Dex in doses 0.004–0.034 mg/kg (Appendix A.1) rarely resulted in genuine SWD appearance (the median of nSWD was 0/per hour with Q1 = 0 and Q3 = 0.138). Differences in the number of SWDs in the Dex condition and control during the full-length EEG were insignificant (Wilcoxon matched pairs test, *p* = 0.65; Appendix A). Figure 3b demonstrates individual data of the density of SWDs at baseline and after Dex injections: the difference between dSWDs of the two conditions was not significant (Wilcoxon matched pairs test, *p* = 0.73). Therefore, Dex did not increase genuine SWDs in non-epileptic control rats.

Two rats (F5 & F6 or 15% of the control group) showed no SWDs at baseline, but some SWDs occurred after Dex injection at dose 0.008 mg/kg (Figure 3a, extreme values in Figure 3b, Appendix A). These subjects were not excluded from the non-epileptic group but were considered as subjects with the latent form of absence epilepsy. Therefore, the alpha2-adrenoreceptor agonist could be considered a drug with the potential to uncover hidden epilepsy.

Noteworthy is that Dex (0.004–0.034 mg/kg) did not provoke genuine SWDs in non-epileptic subjects but elicited epileptiform activity in the form of irregular spikes, 4–8 Hz sharp waves and occasional spike-wave complexes (Appendix A). The power at the central SWDs frequency in 8–10 Hz did not reach the threshold level necessary for detecting genuine SWDs in the baseline (Appendix A) nor in the Dex condition (Appendix A). The abovementioned epileptiform activity did not meet the criteria for SWDs. It might be identified as the rudimentary SWDs in the 4–8 Hz frequency band, which is a lower frequency than in genuine SWDs (rudSWDs in Appendix A).

### 2.2. The Effect of Dexmedetomidine on Genuine SWDs in Rat with Spontaneous Spike-Wave Epilepsy (Epileptic Phenotype)

This part of the study was performed in 13 rats of both sexes (11 females and two males), with the epileptic phenotype confirmed by EEG investigation. This group included 11 WAG/Rij rats, 1 NEW rat and 1 Wistar rat (Appendix A), in which genuine SWD was regularly present in the baseline EEG (Figure 4a). The number of SWDs at baseline varied from 3.4 to 24.2 per hour (12.5 ± 10.8). In addition, the mean duration of SWD varied from 2.91 to 7.67 s (4.96 ± 1.65 s).

Injections of Dex in low doses induced long-lasting SWDs (Figure 4b), which were classified as genuine SWDs based on their time-frequency characteristics (Figure 4b). Pharmacologically induced SWDs and genuine baseline SWDs had similar time-frequency structures and were detected using the same power-frequency parameters (Appendix A). Visual examination of Dex-induced spike-wave activity showed that genuine SWDs were interrupted by periods with epileptiform activity (sharp waves, occasional spikes) that did not meet the criteria for SWDs (Section 2.4). 

Figure 5b shows EEG and power spectra at baseline and after Dex injection. SWDs (marked by white arrows in power) are extremely frequent in Dex conditions. 

The number of SWDs (nSWD), SWDs density (dSWDs, duration of SWDs per min) and the mean duration of SWDs (durSWD) were computed. All these parameters, e.g., nSWD, dSWDs and durSWDs, were statistically analyzed with bin = 5 min and 1 h. In addition, we computed the difference in dSWD values between baseline and Dex conditions in 5 min bins for each rat.

To organize the observed data in meaningful structures, we performed a K-means cluster analysis of SWDs density (bin = 5 min, 6 h). For each 5 min bin, we computed the difference between dSWD measured after Dex injection and in the baseline. The resulting differential SWD scores underwent K-means cluster analysis. We used 50% of the time spent with SWDs as an objective EEG-based criterion for status epilepticus. Four clusters of Dex-induced changes in SWDs density were defined (Figure 5b): (1) ‘Low/no effect’—*n* = 7 rats, Dex dose 0.003−0.049 mg/kg); (2) ‘Pro-absence effect’—9 rats, Dex dose 0.003−0.037 mg/kg; (3) ‘Severe absence, epi-status (early)’—2 rats; Dex dose 0.008−0.012 mg/kg; (4) ‘Severe absence, epi-status (late)’—1 rat with dose two doses of Dex—0.004 and 0.008 mg/kg. Appendix A shows individual data.

Cluster analysis (Figure 5) showed that injections of Dex in doses 0.004−0.012 mg/kg induced severe absence epilepsy with a density of SWDs > 50% from the total time in 3 rats (Clusters 3 and 4 in Figure 5a). Therefore, severe absence epilepsy that lasted longer than 1 h, might be interpreted as epileptic status (epi-status in Figure 5a) with early onset (Cluster 3) and late start (Cluster 4). Furthermore, since SWD density exceeded 50% of the total time in both clusters, clusters 3 and 4 were joined together. In general, K-means cluster classification showed that i.p. injections of Dex affected epileptic activity in rats with spontaneous SWDs in 3 ways (Figure 4): Low effect (Clusters 1);Pro-absence (Clusters 2);Severe epilepsy (Clusters 3 and 4).

Figure 6 demonstrates the dynamics of SWDs density during a 23-h period in all 13 epileptic rats. SWDs density was scored per 1 h during baseline and after systemic administration of Dex (median dose = 0.008 mg/kg; min = 0.003 mg/kg, max = 0.049 mg/kg). The data were normally distributed (K-S test, *p* > 0.05). The RM ANOVA indicated that injections of Dex caused an increase in the density of SWDs compared to baseline (F [1;572] = 9.3, *p* = 0.005). In addition, post-hoc analysis (Duncan test, *p* < 0.05) of the data showed that the density of SWDs was significantly higher than in baseline during 1–5 h after injection. The individual results of SWDs density, which are summarized in Figure 6, can be found in Appendix A.

Figure 7 demonstrates parameters of SWDs assessed in different clusters (categories), such as the ‘low effect’, the ‘pro-absence effect’ and ‘severe epilepsy’ per 1 h during a 23-h period. Rats underwent several tests with different doses of Dex (Figure 2). Therefore, different tests in the same rat might be classified at separate clusters. The data were non-normally distributed and shown as medians and ranges. The ‘low effect’ of Dex with the median dose of 0.010 mg/kg was found in 11 tests on seven rats (Figure 7a). The ‘pro-absence’ effect of Dex was found with the median dose of 0.008 mg/kg in 11 tests on nine rats. The ‘severe epilepsy’ was found after Dex injection with the median dose of 0.008 mg/kg in 3 rats. Interestingly, roughly similar doses of Dex 0.008–0.010 mg/kg differently affected SWDs, as shown by cluster analysis.

The variability of Dex-induced effects on SWDs might be explained by individual factors, such as parameters of SWDs at baseline.

### 2.3. The Retrospective Study of Baseline SWDs in Epileptic Rats

SWDs properties were retrospectively analyzed at baseline considering Dex-induced effects on SWDs: ‘low effect’, ‘pro-absence effect’ and ‘severe epilepsy’ (Figure 6b). Properties of baseline SWDs were analyzed in 23 h of baseline EEG recordings with bin = 1 h using Friedman ANOVA (Figure 6c). The significant effect of the ‘category’ on baseline dSWDs (F[2, 593] = 13.5, *p* = 1.7 × 10^−9^, Figure 6c) suggested that the Dex-induced effect might be predetermined by the density of SWDs in the baseline. In rats with ‘severe epilepsy’, the density of baseline SWDs (2.00 s per hour, Q1 = 1.13, Q3 = 3.39) was significantly higher than in rats with ‘the pro-absence effect’ (1.51 s per hour, Q1 = 0.71, Q3 = 2.69, Mann-Whitney U test, *p* = 2.3 × 10^−3^) and higher than in rats with ‘low effect’ (1.20 s per hour, Q1 = 0.65, Q3 = 1.91, *p* = 6.2 × 10^−8^).

Surprisingly, the number of baseline SWDs did not differ in different categories of Dex-induced effects (F[2, 593] = 1.7, *p* = 0.18, Figure 6c). The median number of SWDs at baseline was 10–11 per hour in all categories.

Duration of baseline SWDs also linked to Dex-induced effects, as was shown by the significant effect of the ‘category’ (F[2, 593] = 46.9, *p* < 1 × 10^−10^, Figure 6c). In rats with ‘severe epilepsy’, the baseline SWDs were longer (6.00 s, Q1 = 5.14, Q3 = 7.27) than in rats with ‘the pro-epileptic effect’ (4.80 s, Q1 = 4.03, Q3 = 5.94, *p* = 8.4 × 10^−11^) and longer than in rats with ‘low effect’ (4.05 s, Q1 = 3.35, Q3 = 5.27, *p* < 1 × 10^−12^).

Our results generally indicated that subjects in which Dex caused severe absence epilepsy had longer SWDs at baseline. However, the pro-absence effect of Dex did not relate to the number of baseline SWDs. Therefore, subjects with long SWDs at baseline were at high risk of absence status after activation of alpha2-ARs.

### 2.4. The Effect of Dexmedetomidine on Vigilance States in Rat with Spontaneous Spike-Wave Epilepsy (Epileptic Phenotype)

Visual analysis of video-EEG records was performed 4 h after injection of Dex using LabChart v. 8.1.16 software. The effects of different Dex doses ranging from 0.003−0.031 mg/kg on EEG and behavioral states were evaluated in 16 EEG records obtained from 6 female rats. Five states were identified based on visual analysis of raw EEG, EEG power-spectrum and video records (where available): (1) wakefulness (Appendix A), (2) slow-wave sleep (Appendix A), (3) spike-wave epileptic activity (Appendix A), (4) deep sedation phase (Appendix A) and (5) the state with mixed EEG patterns (Appendix A).

I.p. injections of Dex usually provoked sedation characterized by specific EEG and behavioral correlates (Appendix A)—light sedation associated with prominent delta waves in EEG mixed with sharp peaks specific to spike-wave epileptic activity. Deep sedation was associated with powerful delta activity without sleep spindles, similar to deep slow-wave sleep [57]. Rats entered a deep sedation phase throughout the state with mixed EEG patterns. The moment of posture loss against the background of a mixed EEG pattern was marked as the beginning of sedation, and the first movement following that was marked as the end of sedation.

The following measures of behavioral/EEG states were analyzed (Figure 8).

The latency of the first SWDs after the injection (L1swd).The latency and the total duration of the sedative state (L2 and D2, respectively)The duration of the deep sedation phase (if present, D3).The duration of the first and the second periods of enhanced spike-wave epileptic activity (D1swd and D2swd, respectively).

Injections of Dex elicited two types of EEG/behavioral responses.

The type 1 response (Figure 8a, Table 2) was characterized by a biphasic increase in spike-wave epileptic activity. It was observed in five rats after eight injections of Dex (doses ranging from 0.0035 to 0.0307 mg/kg). The first phase of elevated spike-wave epileptic activity occurred several minutes after Dex injection and was relatively short (from 68 s to 6.6 min). Then an animal gradually entered a sedative state, i.e., resting posture was lost, but the eyes stayed open; genuine SWDs were replaced by irregular spikes mixed with slow delta waves (so-called ‘state with mixed EEG patterns’). Then a deep sedation phase was observed for 30 s–45 min. During emergence from sedation, irregular sharp spikes appeared again. After that and even before the restoration of muscle tone, genuine SWDs could occur in a desynchronized low-amplitude EEG background for a couple of minutes. The sedative state was terminated with the first body movement after long muscle relaxation. SWD continued to occur during the awake state marking the beginning of the 2nd phase of enhanced spike-wave epileptic activity. The 2nd phase of enhanced spike-wave epileptic activity lasted from 42 min to 2.8 h, which was longer than the 1st phase (Wilcoxon paired test, *p* < 0.01).

The type 2 response to Dex injection (Figure 8b, Table 2) was characterized by a single-phase increase in spike-wave epileptic activity. It was observed in 4 rats after 8 Dex injections (doses ranging from 0.0033 to 0.0120 mg/kg). From 1.5 to 8.5 min after Dex injection, the intensity of spike-wave epileptic EEG activity rapidly increased and was occasionally interrupted by short episodes of wakefulness or by the state with mixed EEG patterns (Figure 7a). No transitions to the deep sedation phase were observed. This period of abnormal epileptic activity lasted for 1–3.9 h and was followed by a normal sleep-wake cycle.

Pearson correlation test was used to determine the dose-specific effect of Dex on the parameters of behavioral/EEG states. Positive correlations were found between the total duration of the Dex-induced sedative state and the dosage of Dex (*n* = 8, r = 0.79, *p* = 0.019, Figure 8a). Positive correlations were also found between the duration of the deep sedation phase and the dosage of Dex (*n* = 8, r = 0.94, *p* = 0.001, Figure 8b). The duration of the 2nd period of enhanced spike-wave epileptic activity (after the sedative state) and the dosage of Dex also correlated positively (*n* = 8, r = 0.71, *p* = 0.045, Figure 9c).

## 3. Discussion

In the present report study, we described in detail the effect of different doses of alpha2-ARs agonist dexmedetomidine on the spike-wave epileptic activity in WAG/Rij rats. The pro-epileptic and the sedative properties of the drug were considered to explain the different reactions to its administration. Besides, we tested its influence on non-epileptic animals.

The control group contained non-epileptic rats of the NEW substrain (Appendix A). In these subjects, spontaneous SWDs were extremely rare (max 0.25/per hour). Around 85% of NEW rats showed no SWDs after Dex administration (0.004–0.034 mg/kg, i.p.). Therefore, Dex did not elicit de novo SWDs in rats without SWDs at baseline. However, in 15% of control rats, some SWDs occurred after Dex injection at 0.008 mg/kg. We assumed that these subjects had the latent form of spike-wave epilepsy, which was disclosed with the pharmacological treatment with the agonist of alpha2-adrenoreceptors. Considering that Dex is regularly used in clinical practice [33,34,37,58], EEG examination in patients using low doses of Dex might help to diagnose the latent forms of absence epilepsy (or pathology of cortico-thalamo-cortical circuitry).

Noteworthy is that Dex (0.004–0.034 mg/kg, i.p.) elicited epileptiform activity in the form of irregular spikes, 4–8 Hz sharp waves and occasional spike-wave complexes in non-epileptic (Appendix A) and in epileptic rats (“State with mixed EEG patterns”, Appendix A). This epileptiform activity did not meet the criteria for SWDs (the central frequency of 8–10 Hz), and some 4–8 Hz events might be called rudimentary SWDs. Classification of the epileptiform activity in Dex conditions might give a clue to disclose hidden pathologies of cortico-thalamo-cortical circuitry.

### 3.1. Spike-Wave Promoting Effect of Alpha2-AR Agonist, Dexmedetomidine

We found that systemic administration of Dex (median dose = 0.008 mg/kg; min = 0.003 mg/kg, max = 0.049 mg/kg) increased the density of SWDs during 1–5 h after injection. Duration of Dex-induced elevation of SWDs could be accounted for by the pharmacokinetics of Dex in rats: the half-life period, T½, is about 1 h [59]. The observed effect was present during the 4 T½–5 T½.

SWD-promoting effect of Dex was mostly linked to the epileptic phenotype of rats with spontaneous SWDs, but seemed not to be dose-dependent:In 23% of tests in rats with spontaneous SWDs (3 out of 11 rats), i.p. injections of Dex with caused the most severe pro-epileptic effect a median dose of 0.008 mg/kg. These rats showed the longest SWDs at baseline (median 6.00 s).A mild pro-epileptic effect of Dex was found in about 46% of rats. These rats had shorter SWD durations at baseline (median 4.80 s) than rats from the previous group.A low effect of Dex on SWDs was found in the rest, 31% of rats. These rats had the shortest SWDs at baseline (median 4.05 s).

A pro-epileptic effect of Dex is associated with the increase of SWDs number rather than with an increase in the duration of SWDs.

The most severe effect of Dex was observed in rats with longer baseline SWDs. However, the pro-epileptic effect of Dex did not relate to the number of baseline SWDs. Therefore, subjects with long SWDs at baseline were at high risk of absence status after activation of alpha2-ARs.

### 3.2. Status Epilepticus Induced by Alpha2-AR Agonists, Dexmedetomidine

We found that relatively low doses of Dex (0.004−0.012 mg/kg, i.p.) resulted in prolonged and recurrent SWDs in 23% of epileptic rats. Severe absence epilepsy that lasted longer than 1 h and might be interpreted as epileptic status [45] with early onset (Figure 4a, Cluster 3) and late-onset (Figure 4a, Cluster 4). Trinka et al. in Epilpesia wrote: “*Status epilepticus is a condition resulting either from the failure of the mechanisms responsible for seizure termination or from the initiation of mechanisms which lead to abnormally prolonged seizures*” [60]. The duration of SWDs was not largely affected by Dex. Therefore, the mechanisms of seizure termination were just slightly altered in epileptic rats. On the other hand, Dex (0.004−0.012 mg/kg, i.p.) elicited numerous SWDs for several hours. Therefore, it might activate mechanisms of seizure initiation.

As we have suggested, low Dex doses led to the achievement of alpha2/alpha1 ARs activation ratio, which was favorable for SWDs prolonged enhancement, but were too low to induce the sedative state. This corresponded to the Type 2 response in Figure 8 and the experimentally refined scheme in Figure 10.

### 3.3. Alpha-Adrenergic Mechanisms of Sedation and Provocation of SWDs

In Dex condition, pharmacologically induced genuine SWDs appeared during active wakefulness, but baseline SWDs in WAG/Rij rats usually occurred during drowsiness or slow-wave sleep [39,40]. This phenomenon was described in our earlier report [22] and at https://encyclopedia.pub/video/video_detail/628 (accessed on 15 April 2023).

The current study observed two changes in EEG/behavioral states after Dex injections. First, a biphasic increase in spike-wave epileptic activity was associated with entering the sedative state, temporarily suppressing the SWDs. Second, Dex dose-dependently induced light or deep sedation. The deep sedation was associated with the loss of righting reflex (LORR) and stable, strong delta activity in EEG in animals [61]. We did not evaluate the righting reflex in the current study, but we used EEG changes to detect the deep sedation phase and intermediate state with mixed EEG patterns corresponding to light sedation. In line with Garrity et al. study [61], we have also observed the dose-dependency of the sedative state and deep sedation phase durations. In type 2 response to Dex injection, a stable sedative state was not achieved, and high spike-wave epileptic activity was dominant. It was interrupted by waking or semisedative states (i.e., a mix of EEG patterns specific to wakefulness and sleep/sedation) for very short periods. The local injection of another alpha2 agonist, clonidine, provoked SWDs without a sedative effect [62]. In our earlier study, we have not noticed a sedative effect of clonidine (0.00625 mg/kg, i.p.) in WAG/Rij rats but observed a severe pro-epileptic effect putatively mediated by the reticular thalamic nucleus [24].

The summary of our results helped add more details to our schema and a pro-epileptic state of “alpha-2 wakefulness” induced by Dex (Figure 9). According to our hypothesis, Dex activates presynaptic alpha2-ARs in LC neurons, thus decreasing LC activity. As a sequence, NA release decreases, and the target cells (including the thalamus) receive less NA modulation (via postsynaptic alpha1- and beta-ARs). In addition to that, Dex might directly activate postsynaptic alpha2-ARs in the thalamus. During natural falling asleep, the LC and NA brain level activity is known to decrease [1,41], but less than after pharmacological stimulation of alpha2-AR. Our previous studies indicated that in Dex conditions, pharmacologically induced genuine SWDs could appear during active wakefulness, even during the feeding process, as shown in [22] and in https://encyclopedia.pub/video/video_detail/628 (accessed on 15 April 2023). At the same time, the baseline SWDs in WAG/Rij rats usually occurred during drowsiness or slow-wave sleep [39]. This led us to the conclusion that the Dex-induced increase of alpha2 activity was more rapid than in the natural state of drowsiness. Therefore, the occurrence of SWDs was shifted from drowsiness to an active wakefulness state. Here, we defined a specific abnormal state favorable for SWDs—“alpha2 wakefulness” This is a paroxysmal wakefulness state that is pharmacologically induced by systemic administration of the agonist of alpha2ARs. "Alpha2 wakefulness" state was dose-specific and individually specific, meaning that different individuals experienced the “alpha2 wakefulness” at different doses of Dex. Moreover, the same doses of Dex induced different reactions in different subjects, therefore, the effects of Dex on EEG/behavioral states were individual. If the dose of Dex was relatively high for a particular rat, the sedative state with delta activity in its EEG occurred. After this sedative state, a rat would enter the second epileptic phase with spike-wave activity, and the ratio of alpha1/alpha2-ARs activity would return to the value typical for wakefulness (Figure 10, Type 1 response). If the dose of Dex was relatively low for a particular rat, an epileptic phase with spike-wave activity occurred, and no sedation happened (Figure 10, Type 2 response).

Buzsáki et al. 1991 demonstrated that HVS epileptic activity (same as SWDs) in rats could be provoked by activating the specific pool of alpha2 postsynaptic ARs in the thalamus [21]. The remarkable fact is that the alpha2b subtype of ARs is placed almost exclusively in the thalamus [63]. Therefore alpha2b-AR seems to be a possible molecular target for selective therapy of spike-wave absence epilepsy [22].

### 3.4. Limitations and Further Directions

The lack of sham injections as a control condition is a limitation of the current study. Here we used baseline records for control but not records after control injections. A similar approach was chosen in our previous study on the effect i.p., injections of another alpha2-agonist—clonidine, in which baseline was used as a control condition [24]. In the preliminary stage of the current study, we examined the effect of a sham injection in one epileptic rat. We did not observe the SWDs pattern specific to Dex response of any type: a consequence of SWDs separated from each other just by short breaks several minutes after the injection. Also, in the present research, we examined the effect of Dex on SWDs’ number and duration in different clusters, and these data were obtained at the same Dex condition.

We have proposed that lower doses of Dex would lead to long and stable spike-wave epileptic activity because the rats were “trapped” in the “alpha2 wakefulness state” favorable for the occurrence of SWDs without entering the sedative state. Nevertheless, a type 2 response to Dex could be observed at the same doses (0.0035–0.0307 mg/kg, i.p.) as a type 1 response (0.0033–0.0120 mg/kg, i.p.). In the current study, different rats had different baseline levels of SWDs and probably different sensitivity to Dex. Therefore a dose-dependent effect considering the type of response was not found. Perhaps a careful investigation of individual dose-dependency could help to define the switch of response type from single phase to biphasic from low to high dose of Dex. Nevertheless, in the type 1 response, the dose-dependent increase of the second phase of enhanced spike-wave epileptic activity duration was demonstrated.

## 4. Materials and Methods

The study was conducted on adult WAG/Rij rats and the non-epileptic control “NEW” rat substrain. All rats were bred and maintained at the Institute of Higher Nervous Activity and Neurophysiology of RAS, Moscow (IHNA). The experiments were carried out in accordance with EU Directive 2010/63/EU for animal experiments and were approved by the animal ethics committee of IHNA (protocol No. 4 was approved on 26 October 2021). Rats were kept in environmentally controlled conditions with a 12:12 h light: dark cycle (lights on at 8 a.m.), with constant ventilation and airing. Rats were provided with unlimited access to food and water. Rats were housed in same-sex groups (3–4 subjects per cage) to minimize any potential stressors and to promote social interactions.

In total, 26 adult rats of both sexes were used (incl. 13 subjects with the epileptic phenotype—11 WAG/Rij, 1 Wistar, 1 NEW; and 13 subjects with the non-epileptic phenotype—NEW rats). The average body weight of females was 261 g (min 230 g; max 355 g), and 385 g for males (min 340 g; max 446 g).

### 4.1. Implantation of Chronic EEG Electrodes, EEG Recording and Reduction of the Potential Biases

Chronic electrodes were implanted to record electrical brain activity in free behavior. Epidural electrodes were made of stainless-steel screws (screw: shaft length = 2.0 mm, head diameter = 2.0 mm, shaft diameter = 0.8 mm), providing recordings of electrical brain activity (EEG). The surgery was performed under isoflurane anesthesia in a stereotaxic apparatus (Kopf model 900, David Kopf Instruments, USA). Three active epidural screw electrodes were implanted over the left/right frontal cortex (AP ± 2; L 2.5) and occipital cortex (AP − 6; L 3); the fourth reference screw electrode was placed over the right cerebellum. Coordinates are given in mm relative to the bregma. Care was taken to reduce pain, suffering, and distress as indicated in the experimental protocol approved by the animal ethics committee of IHNA RAS. After the surgery, animals were housed individually under a 12:12 h light: dark cycle (light on 8 a.m.) with free access to food and tap water. The recovery period lasted 10–14 days.

Rats were placed in Plexiglas cages (25 cm × 60 cm × 60 cm), and three-channel EEG signals were recorded in freely moving animals under a 12:12 h light: dark cycle (light on at 8 a.m.). Signal inputs were fed into a multi-channel amplifier (PowerLab 8/35,model PL3508 with 8 bridge Amp model FE221, ADInstruments, Sydney, Australia) via a swivel connector, band-pass filtered between 0.5 and 200 Hz, digitized with 400 samples/second/per channel, and stored in a hard disk. After Dex injections, eFace 1325R video camera (Genius, Taiwan) was used to record the rat’s behavior for 2–4 h.

No animals were harmed or lost during experiments because Dex was used in therapeutic doses. All rats were born, raised, and given all the same conditions in the same environment. After the surgery, the rats were allowed to recover for ten days and kept in individual cages on the same two lower rows. Experiments were performed in the sound-attenuated room with the same characteristics as in the vivarium. We also performed baseline recording before each Dex injection, which helped reduce some potential biases.

### 4.2. Drugs

All pharmacological tests were done using the agonist of alpha2-adrenoreceptors dexmedetomidine hydrochloride (Dexdomitor^®^ 0.1, Orion Pharma, Espoo, Finland). Dex was injected i.p. in different therapeutic doses ranging from 0.003 to 0.049 mg/kg. To select the most effective dose with the highest pro-epileptic effect, i.e., the central dose, we did pilot experiments in epileptic rats. The effect of Dex was individual, and the doses that were given varied across the central dose. Non-epileptic rats were injected with the same (central) dose of Dex.

### 4.3. Analysis of EEG Data

Three-channels EEG recordings were visually analyzed in time and frequency domains (LabChart 8 version 8.1.16, ADInstruments). The preliminary frequency analysis of video-EEG was performed using FTT with window sizes 512, 1024 or 2048 and 50–95.5% window overlap (Appendix A).

#### 4.3.1. EEG-Based Visual Detection of Behavioral States

Visual analysis of video-EEG records was performed 4 h after injection of Dex using LabChart 8 software (v. 8.1.16, ADInstruments). The frontal left EEG channel underwent power spectrum analysis (FFT size 1024, Hann (cosine bell), window overlap 50%). Based on raw EEG, EEG power-spectrum and video records (when available), five states were identified: (1) wakefulness, (2) slow-wave sleep, (3) spike-wave activity, (4) deep sedation phase and (5) the state with mixed EEG patterns. To evaluate the whole EEG-behavioral picture of rats’ reaction to Dex administration, video-EEG was recorded in each rat used to assess EEG/behavioral states. We checked whether we could use pure EEG data to identify EEG/behavioral states. The preliminary analysis showed that this approach was reliable.

Wakefulness (Appendix A). This state was associated with body movements (large or small amplitude) or still posture while preparing to fall asleep. The frontal left EEG during wakefulness showed low amplitude (around 300 uV) and fast activity. The power density spectrum had two dominating frequency bands: delta (0.5–4 Hz) and theta (around 7 Hz). Wakefulness also included feeding periods (active wakefulness) when specific artifacts were present in the EEG (see Appendix A for illustrations).

Slow-wave sleep (Appendix A) is characterized by closed eyes and a specific sleeping posture with a particular muscle tone. The frontal left EEG showed mostly slow, high-amplitude activity (up to 1000 uV) at 0.5–4 Hz, with some 8–12 Hz sleep spindles mixed in.

The spike-wave epileptic activity (Appendix A) was associated with behavioral arrest and immobility. High-voltage (1200−2000 uV) repetitive spikes at 7−8 Hz were present in the frontal left EEG. This epileptic activity was also automatically identified as spike-wave discharges, SWDs (Appendix A). Dex injections enhanced spike-wave activity with numerous discontinuous SWDs interrupted by non-SWD activity lasting from several seconds to several minutes. For the analysis of behavioral state dynamics, long periods containing SWDs were marked as ‘spike-wave epileptic activity’ if time intervals between successive SWDs were less than 5 min.

Deep sedation phase (Appendix A). Periods with very low muscle tonus (i.e., the rat was lying flat on the stomach) and eyes opened. The EEG showed mostly slow (0.5−4 Hz), high-amplitude activity (up to 1000 uV). There was just slight modulation of the frontal left EEG signal amplitude.

State with mixed EEG patterns (Appendix A). Periods showing mixed behavioral and EEG patterns of deep sedation phase and spike-wave epileptic activity: very low muscle tonus (animal is lying flat on its stomach) and eyes opened, slow delta waves predominance in the frontal left EEG intermittent with the epileptiform activity (spikes and sharp waves, but not mature SWDs).

#### 4.3.2. EEG Analysis of Epileptic Activity in the Form of Genuine Spike-Wave Discharges

SWDs were detected based on the criteria introduced in the classic paper of G. van Luijtelaar and A. Coenen in 1986 [64]: “*The spike-wave complexes are asymmetric, the repetition of spikes within a burst varies from 7.5 to 9.5 Hz with a mean frequency of 8.7 Hz.*” However, the majority of pharmacological studies in WAG/Rij rats were done at the University of Catanzaro (Italy), and Marmara University (Istanbul, Turkey) still define SWDs manually (for example, [65,66,67]). At the same time, several groups agreed that the continuous wavelet transform is the best way to detect the SWD [56,68,69,70,71]. Therefore, our group developed the method for the automatic detection of SWDs based on the continuous wavelet transform with the complex Morlet wavelet [56,72]. See more details in Appendix A. This automatic method of SWDs detection has the following advantages: (1) it is faster and more efficient than the manual method of detection; (2) it is more accurate and reliable than the manual method of detection.

In the current study, SWDs were automatically detected in the full-length raw EEG using the wavelet-based detection algorithm (see [68] for technical details). An increased wavelet power at the characteristic frequency of 8–10 Hz and 17–20 Hz (1st harmonic) was used as a distinctive feature of SWDs. The full-length EEG was processed by continuous wavelet transform with the complex Morlet basic function. Wavelet power was computed in the main frequency band of SWDs, [8–10 Hz], and [17–20 Hz] (Appendix A). Each band’s threshold values of wavelet power were chosen individually by testing different values and seeing which produced the most accurate results. SWDs were detected when wavelet power in both bands exceeded threshold values. To improve the method’s selectivity, the seizure activity’s minimal duration was set at 2 s. The automatic detection results were checked by an expert and corrected if necessary.

The density of SWDs was scored as the sum of SWD duration per second in bins 5 min and 1 h. The number of SWDs was average per 1 h. The mean duration of SWDs was scored as average per 1 h.

### 4.4. Estral Cycle

The oestrous cycle phase in females was determined by direct cytology in wet smears immediately after collection using an unstained slide [20,43,44]. Vaginal swabs were obtained using sterile saline, and the cells were examined under a microscope in a drop of saline. Microscopic examination was performed using a Nikon microscope (ECLIPSE E200, Nikon Instruments Inc, Japan) with a 10× objective to determine the relationship between cell types and a 40× objective to recognize cell types.

Diestrus (the longest phase) lasted, on average, 48–72 h and was characterized by the predominance of leukocytes. Leukocytes were absent in the stage of proestrus and estrus. However, some leukocytes and epithelial cells were detected during the metestrus (6–8 h).

### 4.5. Statistical Analysis

Datasets containing the start and end points of automatically detected SWDs were statistically analyzed. First, the densities of SWDs were computed per 5 min as the total duration of SWDs in seconds during 5 min. Second, the number of SWDs, duration of SWDs and density of SWDs were computed per 1 h. Third, the number of SWDs (nSWD), SWDs density (dSWDs, duration of SWDs per min) and the mean duration of SWDs (durSWD) were computed with bin = 5 min and 1 h. Finally, we computed the difference in dSWD values for each rat between baseline and Dex conditions with 5 min bins during the first 6 h after injection. These differential dSWD values were further used for K-means cluster analysis.

We used Statistica version 12 to perform statistical analysis of the obtained data. In the first step, all the samples were tested for normality. Next, the normal distribution was assessed with a Kolmogorov-Smirnov test with *p* > 0.05. Next, normally distributed data were analyzed with the repeated measures (RM) ANOVA and post-hoc Duncan test.

For samples that did not show Gaussian distribution or small sample size (*n* < 10), we used non-parametric statistical criteria, such as Friedman ANOVA, Mann Whitney U test, Wilcoxon matched pairs test and Pearson correlation test. The level of significance was set at *p* < 0.05.

## 5. Conclusions

In the present study, we evaluated adrenergic mechanisms of generalized spike-wave epileptic discharges (SWDs), which are the encephalographic hallmarks of idiopathic generalized epilepsies. We create the concept of alpha1- and alpha2-ARs regulation of SWDs via NA modulation of cortico-thalamo-cortical network activity. We provided experimental evidence to favour this concept and defined the abnormal state favorable for SWDs to occur—“alpha2 wakefulness”.

Our experimental study was performed in rats with spontaneous SWDs (WAG/Rij rats and Wistar) and in non-epileptic control strain NEW. Activation of alpha2-ARs caused severe spike-wave epilepsy up to a status epilepticus in WAG/Rij rats, but not in non-epileptic control. Although in two non-epileptic subjects of NEW substrain, genuine SWDs were recorded after injection of Dex. The NEW substrain has a non-epileptic phenotype, but it has a genetic predisposition to spike-wave epilepsy. Therefore, giving Dex to subjects with a genetic risk for seizures could make them more likely to have seizures.Furthermore, considering that Dex is regularly used in clinical practice, EEG examination in patients using low doses of Dex might help to diagnose the latent forms of absence epilepsy (or pathology of cortico-thalamo-cortical circuitry).

In epileptic WAG/Rij rats, a pro-epileptic effect of Dex was associated with the increase in the number of SWDs rather than with the increase in duration of SWDs. The most severe impact of Dex was observed in rats with longer baseline SWDs. The pro-epileptic effect of Dex did not relate to the number of baseline SWDs. Relatively low doses of Dex (0.004−0.012 mg/kg, i.p.) resulted in prolonged and recurrent SWDs in 23% of epileptic rats that might be interpreted as epileptic status. In all, subjects with long SWDs at baseline were at high risk of absence status after activation of alpha2-adrenoreceptors.

We observed two types of changes in the rats’ EEG/behavioral state after the Dex injection. First, a biphasic increase in spike-wave epileptic activity: an initial increase in SWDs soon after Dex injection was followed by the sedative state (when SWDs were suppressed) and then by the secondary increase in SWDs. Second, light or deep sedation dose-dependently induced by Dex.

## Figures and Tables

**Figure 1 ijms-24-09445-f001:**
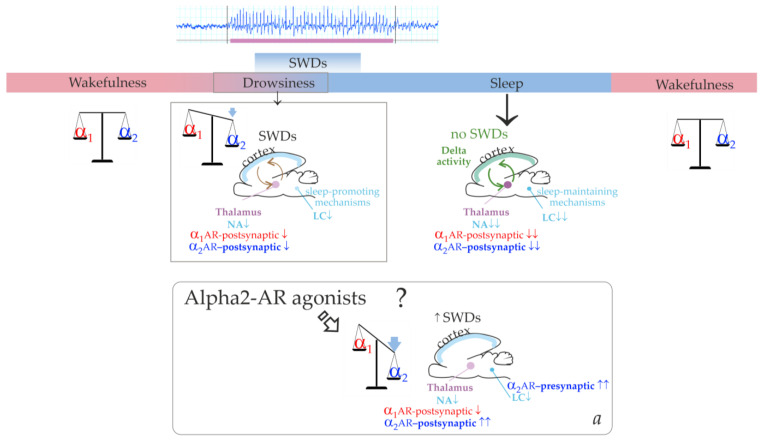
The schema demonstrating involvement of the alpha1- and alpha2-adrenoreceptors (ARs, shown in red and blue font colors respectively) in control of epileptic spike-wave discharges (SWDs), wakefulness, drowsiness and sleep. Locus Coeruleus (LC) is the center of the brain’s noradrenergic system is shown in cyan color. The LC releases and distributes noradrenaline (NA, cyan color). Thalamus is shown in purple color. During wakefulness, alpha1- and alpha2-ARs activation impacts the overall NA effect in a state of dynamic equilibrium. During drowsiness, the firing activity of LC neurons and the release of NA are decreased as cyan font colors and cyan arrows (under the influence of sleep-maintaining mechanisms). Downregulation of thalamic alpha2- and alpha1-ARs (down arrows) increases regular oscillatory activity in the thalamocortical network (circle arrows. During drowsiness and light sleep, the thalamocortical network in genetically prone subjects generates epileptic spike-wave discharges (SWDs). During sleep, thalamic alpha2- and alpha1-ARs become even more downregulated; the cortico-thalamo-cortical network produces slow-wave activity. Delta activity (green font) is generated the thalamocortical network during sleep (circle arrows). Spontaneous SWDs do not occur during sleep (green font). The question mark highlights a hypothetical schema about SWDs-promoting effect of agonists of alpha2-ARs (see description in the text).

**Figure 2 ijms-24-09445-f002:**
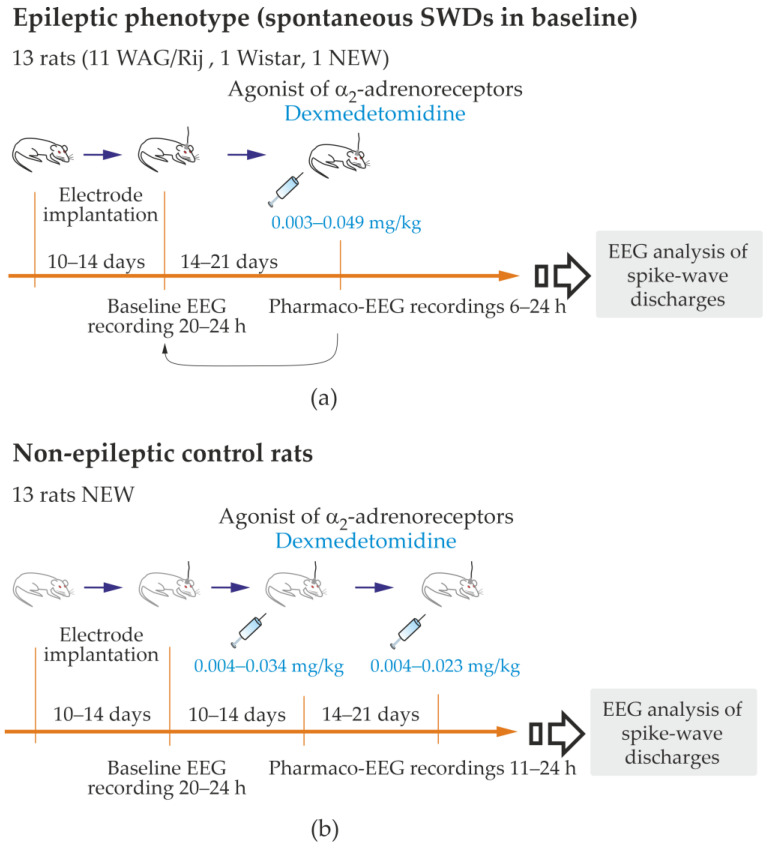
Experimental design of the study. Pro-Epileptic and sedative effects of Dexmedetomidine were examined in rats with spontaneous pike-wave discharges (SWDs), i.e., the epileptic phenotype; (**a**) and in non-epileptic control rats (**b**).

**Figure 3 ijms-24-09445-f003:**
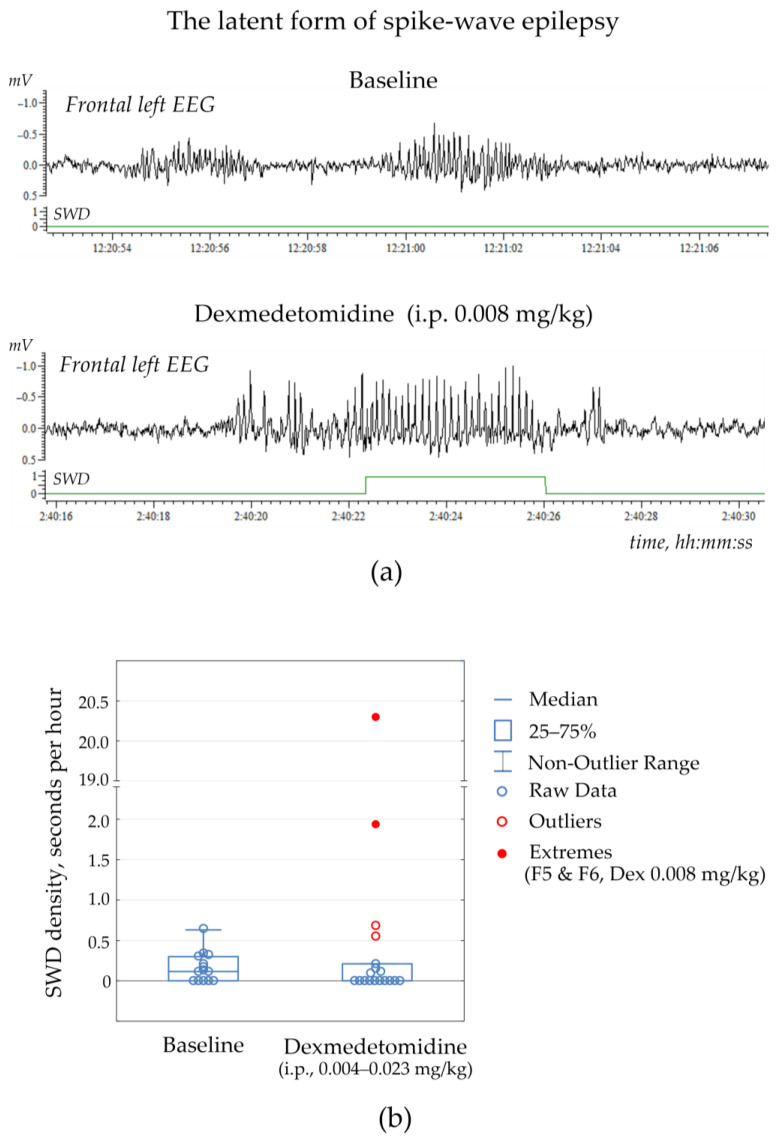
Spike-wave discharges (SWDs) in non-epileptic rats NEW. (**a**) An example of the latent form of spike-wave epilepsy in rat ID = F5, which showed no SWDs in baseline. Systemic administration of Dexmedetomidine (0.008 mg/kg) elicited a genuine SWD 2 h 40 min after the injection. The bottom graph shows automatically recognized SWD. (**b**) Statistical results of SWD density (seconds per hour) as measured in the full-length EEGs in the group of NEW rats (*n* = 13, Appendix A). Two extreme values (red circles) corresponded to subjects with no SWDs at baseline and some SWDs after Dexmedetomidine injection (the “latent” form of spike-wave epilepsy).

**Figure 4 ijms-24-09445-f004:**
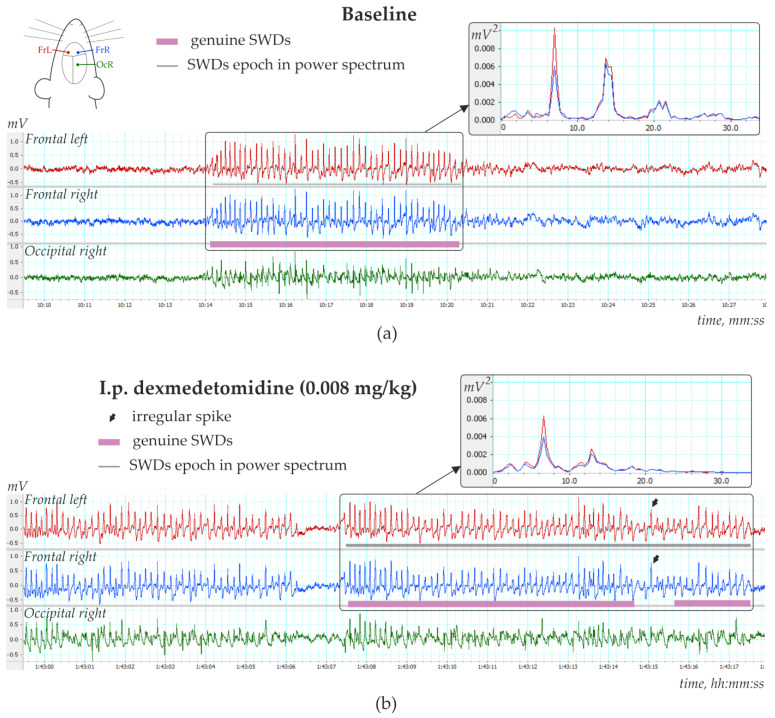
Three-channels EEG with spike-wave discharges (SWDs) recorded in 6-month-old female WAG/Rij rat (**a**) during baseline and (**b**) 1 h 43 min after i.p. injection of dexmedetomidine (0.008 mg/kg). Power spectra are shown in insertions.

**Figure 5 ijms-24-09445-f005:**
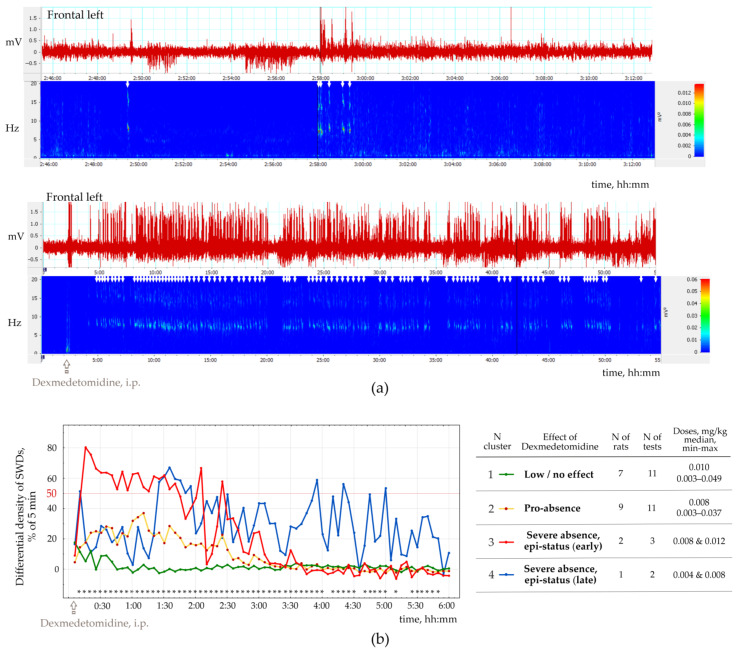
The pro-epileptic effect of alpha2-agonist Dexmedetomidine in rats with spontaneous SWDs. (**a**) Example of SWDs and power spectrum recorded during baseline and 5 min after i.p. injection of Dexmedetomidine (0.011 mg/kg) in rat ID = W7. The frontal left EEG signal was visualized in LabChart v. 8.1.160. The bottom graph shows the FFT power spectrum computed with 1024 size and 95.5% window overlap. SWDs (white arrows) are seen as high-voltage bursts in raw EEG and 8–10 Hz activity in the power spectrum. (**b**) Scores of the differential density of SWDs (the difference between dSWD after Dex injection and in baseline in% from 5 min) in 13 rats (27 tests, 1–3 tests per rat). Results of K-means cluster analysis; asterisks mark 5-min intervals, in which differences between clusters were significant with *p* < 0.05.

**Figure 6 ijms-24-09445-f006:**
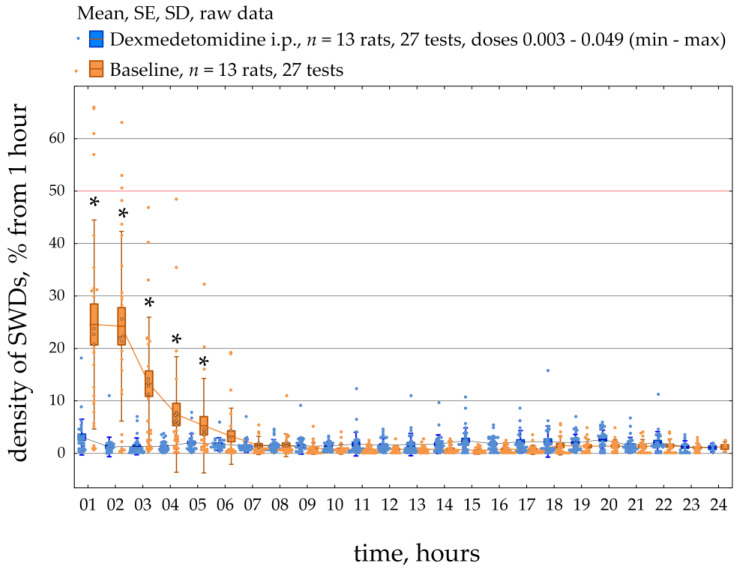
The density of spike-wave discharges (SWDs) is measured in 1 h and presented in percentages from the entire epoch. The data was obtained in the group of 13 epileptic rats in a control condition (orange) and after Dex (blue) administration. *—significant differences according to post-hoc analysis (Duncan test, *p* < 0.05).

**Figure 7 ijms-24-09445-f007:**
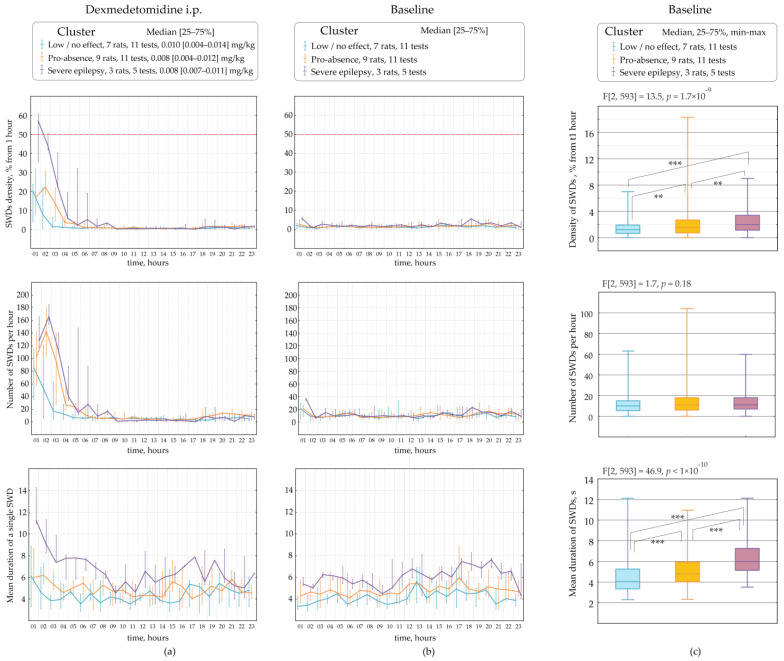
Parameters of spike-wave discharges (SWDs) as measured in epileptic rats (*n* = 13 rats) after Dex injections (**a**) and in baseline (**b**,**c**). The measurements were performed during a 23-h period with a bin size of 1 h. The data were non-normally distributed (Kolmogorov-Smirnov test, *p* < 0.05) and presented as medians and ranges. **—significant differences according to Mann-Whitney U test (*p* < 1 × 10^−3^); ***—*p* < 1 × 10^−5^.

**Figure 8 ijms-24-09445-f008:**
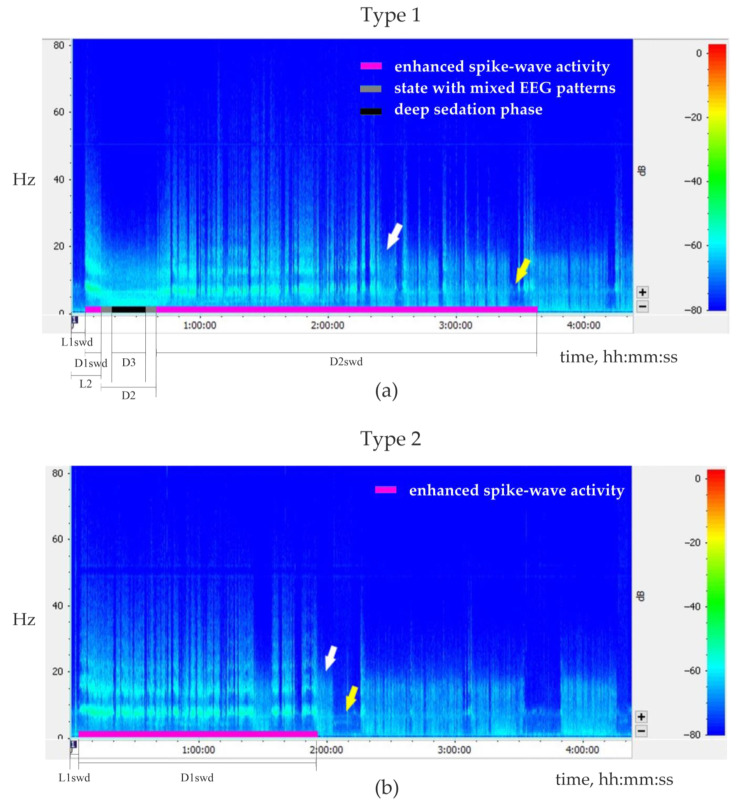
Representative examples of 4-h sonograms for two types of behavioral/EEG responses to injection of Dexmedetomidine (Dex). Plots demonstrate sonograms of the left frontal electroencephalographic signal (FFT size 1024, Hann (cosine bell), window overlap 50%). White and yellow arrows show examples of periods of slow-wave sleep and wakefulness, respectively. Normal sleep-wake cycle starts after the end of enhanced spike-wave epileptic activity. (**a**) The type 1 response: biphasic increase of spike-wave epileptic activity after Dex injection (dose 0.011 mg/kg). (**b**). The type 2 response: one-phase increase of spike-wave epileptic activity after Dex injection (dose 0.004 mg/kg). The following parameters were measured: the latency of the first SWDs after Dex injection (L1swd); for the sedative state: the latency and the total duration (L2 and D2 respectively); for the deep sedation phase: the duration (D3); for the 1st and the 2nd periods of enhanced spike-wave epileptic activity: duration (D1swd and D2swd respectively).

**Figure 9 ijms-24-09445-f009:**
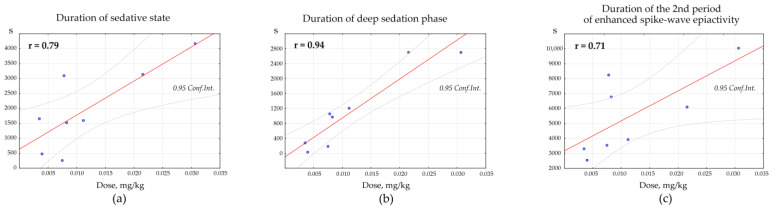
The dose-specific effect of Dexmedetomidine (Dex) on parameters of behavioral/EEG states. The plots demonstrate significant positive correlations between variables (Pearson’s r with *p* < 0.05): the dosage of Dex and (**a**) duration of sedative state (combined deep sedation phase and intermediate periods of state with mixed EEG patterns); (**b**) duration of deep sedation phase; (**c**) duration of the 2nd period of enhanced spike-wave epileptic activity.

**Figure 10 ijms-24-09445-f010:**
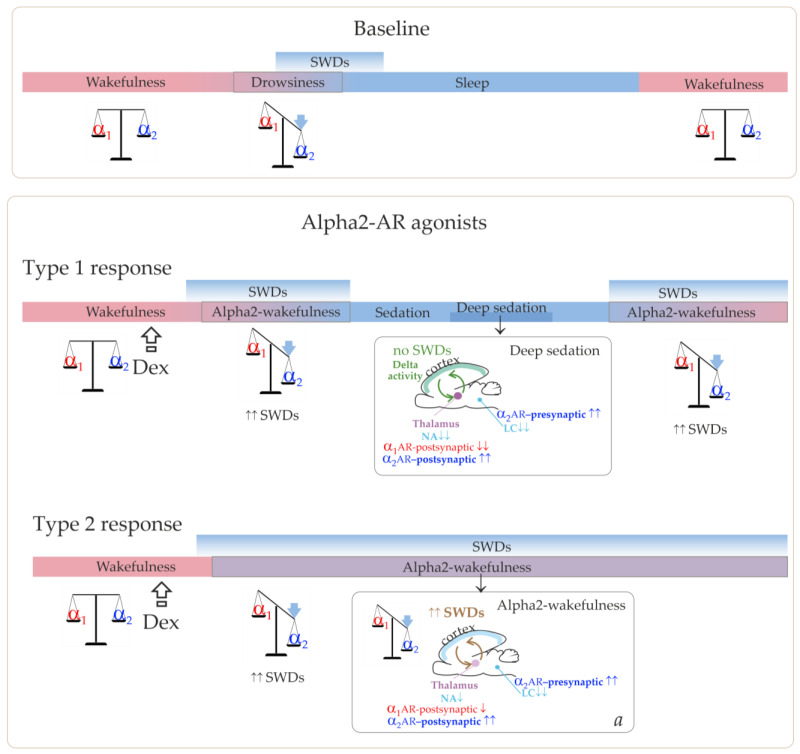
The schema demonstrating adrenergic control of sleep and epileptic spike-wave discharges (SWDs) and SWDs-promoting effect of agonists of alpha2-ARs. Alpha1 and alpha2 adrenoreceptors (ARs) are shown in red and blue colors respectively. Neurons of Locus Coeruleus (LC) releases noradrenaline (NA) and distributes it all over the brain (cyan color). Thalamus is shown in purple color. SWDs do not occur during deep sedation (green font). Our hypothesis about SWDs-promoting effect of agonists of alpha2-ARs (**a**) received experimental proofs. We came up with the term "alpha2-wakefulness" to describe the state with the predominant effect of alpha2 ARs. See description in the text.

**Table 1 ijms-24-09445-t001:** The literature data on alpha2-agonist dexmedetomidine (Dex) effects on spike-wave discharges (SWDs) in rats.

Rout	Dose	Effect	Rats	Reference
i.p.	1 mg/kg, acute	Injection of Dex in a very high dose (1 mg/kg) elicited: Decrease in the total number of SWDs during 40–120 min after injection. No differences in the mean duration of SWDs	WAG/Rij rats 6–7 months old, male	Al-Gailani et al., 2022 [38]
i.c.v.	0.1, 0.5, 2.5 µg, acute	Injection of 2.5 µg led to an enhancement of SWDs: Increase in the total duration of SWDs on 20, 40 and 60 min after treatment. Increase in the number of SWDs on 60 and 80 min after treatment. Increase in the mean duration of SWDs on 40 and 60 min after treatment. Injection of 0.5 µg led to an enhancement of SWDs: Increase in the total duration of SWD 20 min after treatment. Increase in the mean duration of SWD 20 min after treatment	GAERS 3–4 months old, male	Yavuz et al., 2022 [23]
i.p.	0.005 mg/kg, acute	Injection of a very low dose (0.005 mg/kg) increased the incidence of HVS 2 h after injection.	Wistar 10–12 months old, male	Yavich et al., 1994 [26]

Abbreviations: HVS—high voltage spindles (similar to spike-wave discharges), SWDs—spike-wave discharges; GAERS—genetic absence epilepsy rats from Strasbourg. Type of administration: i.p.—intraperitoneal, i.c.v.—intracerebroventricular.

**Table 2 ijms-24-09445-t002:** Parameters of behavioral/EEG states induced by i.p. administration of Dex for Type 1 and Type 2 responses (16 records obtained from 6 rats; doses ranging from 0.0033 to 0.0120 mg/kg). Data are given in median, 1st (Q1) and 3rd (Q3) quartiles.

Number of Records			L1swd, s	D1swd, s	L2, s	D2, s	D3, s	D2swd, s
Type 1	8	Median Q1–Q3	160.5 102.3−316.8	330.0 146.3–590.5	514.0 262.3–854.3	1618.5 1255.0–3103.3	1005.0 252.0–1575.0	4990.0 3482.5–7146.8
Type 2	8	Median Q1–Q3	188.0 168.3–225.5	9257.5 4557.0–12,220.5	-	-	-	-

Abbreviations: L1swd—the latency of the first SWD after the injection; D1swd—the duration of the first period of enhanced spike-wave epileptic activity; L2—the latency of the sedative state; D2—the duration of sedative state; D3—the duration of the deep sedation phase; D2swd—the duration of the second period of enhanced spike-wave epileptic activity.

## Data Availability

All experimental data obtained in the current study are shown in figures and tables. Primary datasets are available from the corresponding author upon reasonable request.

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
