# Peer review of "Alpha2 Adrenergic Modulation of Spike-Wave Epilepsy: Experimental Study of Pro-Epileptic and Sedative Effects of Dexmedetomidine"

_ijms, 2023, doi:10.3390/ijms24119445_

Round 1

Reviewer 1 Report

This is an interesting study by Sitnikova et al. investigating the effects of an alpha2-adrenergic receptor agonist dexmedetomidine (Dex) on generalized spike-wave epileptic discharges in rats. Overall, the study is well conducted, but there are several major concerns that need to be addressed:

The text of the manuscript and English usage need to be significantly improved, and many parts must be rewritten. The current version is at times practically incomprehensible, and it contains an enormous number of typos. Using a native speaker to make appropriate changes is highly recommended.

Introduction is extensive, and it lacks focus, without clearly identifying the gap in knowledge and appropriately formulating the hypothesis to be tested.

The quality of figures needs to be significantly improved. Most of the presented raw EEG data are directly copy-pasted from Lab Chart, with relatively low resolution.

As stated in Page 4, the study was “based on the core idea that pharmacological stimulation of alpha2-adrenergic receptors would trigger two fundamentally different mechanisms”, but these mechanisms mentioned in the same paragraph were not investigated.

How was the Dex dose range determined? Why is the dose range different in epileptic and control rats?

How was the sample size determined.

The analysis of different behavioral states needs to be explained in detail.

The quality of English usage is very poor, and large parts of the manuscript have to be revised.

Author Response

Dear Reviewer,

We are extremely grateful for the time and effort that you dedicated to reviewing our paper. We are very thankful for your suggestions and feedback. Your comments were about the methodology, presentation and clarity of our ideas. Please find our point-to-point answers below.

The text of the manuscript and English usage need to be significantly improved, and many parts must be rewritten. The current version is at times practically incomprehensible, and it contains an enormous number of typos. Using a native speaker to make appropriate changes is highly recommended.

Response: We are sincerely sorry for submitting the first version of our manuscript to the journal without going through a rigorous proofreading process. In the revised version, we made a number of improvements to the text's readability and grammatical accuracy, including fixing spelling and grammar errors.

Introduction is extensive, and it lacks focus, without clearly identifying the gap in knowledge and appropriately formulating the hypothesis to be tested.

Response: We have significantly overworked the introduction.

The quality of figures needs to be significantly improved. Most of the presented raw EEG data are directly copy-pasted from Lab Chart, with relatively low resolution.

Response: We have revised and updated figures 5 and 7.
We agree with critical remarks about the resolution of copy-pasted blocks. We would like to demonstrate the raw data copied directly from LabChart, therefore we did not use any external software to edit the original screenshot. Therefore, these data are presented at a resolution of 300-500 dpi.
This resolution is acceptable for the IJMS and suitable for the online version of the manuscript.

As stated in Page 4, the study was “based on the core idea that pharmacological stimulation of alpha2-adrenergic receptors would trigger two fundamentally different mechanisms”, but these mechanisms mentioned in the same paragraph were not investigated.

Response: We apologize for not clearly expressing our reasoning. The introduction has been revised.

How was the Dex dose range determined? Why is the dose range different in epileptic and control rats?

Response: Dex was used in therapeutic doses recommended in drug annotations. We also considered literature data shown in Table 1. Dex was used in therapeutic doses recommended in drug annotations. We also considered literature data shown in Table 1. We addressed this issue in Section 4.2: "Dex was injected i.p. in different therapeutic doses ranging from 0.003 to 0.049 mg/kg. In order to select the most effective dose that had the highest pro-epileptic effect, i.e., the central dose, we did pilot experiments in epileptic rats. The effect of Dex was individual, and the doses that were given varied across the central dose. Non-epileptic rats were injected with the same (central) dose of Dex."

How was the sample size determined?

Response: In the current study we investigated the effect of different i.p. doses of dexdomitor on epileptic spike-wave activity in WAG/Rij rats and in a new non-epileptic substrain (NEW). This kind of study has not been done before. Inasmuch as no previously obtained data were available, it was not possible to assume the effect of sample size and the standard deviation. IIn this case, the “resource equation” method can be used to calculate the E value. After the clusterization of the SWDs scores in the epileptic group, we finally defined 3 clusters for 13 rats. Thus, E=13-3=10. The corresponding E-value for non-epileptic rats was E=13-1=12. The E-value between 10 and 20 indicates that the sample was not too big and not too small.

The analysis of different behavioral states needs to be explained in detail.

Response: The detailed description of behavioral-EEG states was copied from the Supplementary 4 to the Methods section. The illustrative materials can be found in ‌Supplementary 4.

Reviewer 2 Report

Extensive English editing required.

Author Response

Dear Reviewer,

Thank you for your positive feedback and for highlighting the weak points of our study. We highly appreciate your efforts and time in evaluating our paper. Thank you for your suggestions on ways to report statistical outcomes. Your thoughtful and well-reasoned suggestions helped us to come up with a comprehensive, reliable report. We revised our review in accordance with your critical remarks.

Firstly, I would like to congratulate the authors on their interesting work. However, some points need to be addressed: 

1) It is advisable that the authors include the study design type within the title. 

Response: We have changed the title to “Alpha2 adrenergic modulation of spike-wave epilepsy: the experimental study of pro-epileptic and sedative effects of dexmedetomidine.”

2) The authors should work with English editing services to improve readability of the text. 

Response:  We are sincerely sorry for submitting the first version of our manuscript to the journal without going through a rigorous proofreading process. In order to improve the readability and grammatical accuracy of the text, we used an English editing service.

3) The authors should review the text to remove spelling errors such as sentences beginning with lowercase letters. 

Response: The English was corrected.

4) The authors should review all the abbreviations of the text since there are some which are not previously described before being mentioned, for example in the abstract. 

Response:  This point is taken. In the revised text, we have included the abbreviation list. The abstract is corrected. Abbreviation WAG/Rij rats is a common abbreviation of the rat strain.

5) In the first paragraph of the introduction, I advise the authors to use the term “seizures” instead of “convulsions”. 

Response: Thank you for this remark. The introduction has now been corrected.

6) On page 2, in the text right after table 1, I suggest the authors revise this part and include references accordingly as it seems there are missing citations. 

Response: We have significantly revised the introduction and added citations in order to provide the most accurate and comprehensive information.

7) Self-citations should only be included when essential and expressions such as “(see also our review [22]).” should be removed (first paragraph Page 3). The authors could provide some background such as “In our previous study(…)”, but it is preferable to avoid structures like references and “[]” between parenthesis. 

Response: The reference to ‌item 22 (our review) is necessary, because our concept was first introduced in this peer-reviewed publication. We changed the way we cite sources.

8) Please revise terms such as on page 3 “abovementeined review”. 

Response: Ok.

9) The authors should avoid bullet points and lists along the tests such as in page 11. 

Response: We removed bullet points on page 11 and in Table 1. We prefer to keep bullet points in Sections 2.4 and 3.1, because bullet lists are recommended in the template for comprehension. They make it easier for readers to grasp the main points of an essay.

10) Graphs should have their scales adjusted and tables should be significantly improved. Table 2 on page 13 for example, has spelling and configuration errors. Legends of tables and graphs should also be corrected. 

Response: Tables 1 and 2  were revised as well as their legends. Graphs and figure legends were also corrected.

11) Regarding the study protocol, was this study based on a previous protocol? How were timeline for injections decided? Provide a statement indicating whether a protocol (including the research question, key design features, and analysis plan) was prepared before the study, and if and where this protocol was registered. 

Response:  The study protocol was reviewed and approved by the ethics committee of our institute in accordance with the declaration of Helsinki. This protocol was prepared before the study (https://www.ihna.ru/institute/ethic/0413122022_ethic_protocol.pdf). Details are written in Russian. 

The original protocol in Russian has been uploaded to Google-doc https://docs.google.com/document/d/14zDIje5xmhBPueAxo7QszLztJRSYjteK/edit?usp=sharing&ouid=100065682202293471689&rtpof=true&sd=true

This protocol included Aims/tasks/research questions/hypotheses.

Tasks. 1. Selection and breeding of the non-epileptic WAG/Rij substrain.

  1. Examine changes in the electrical activity of the brain under the influence of pharmacological substances used in veterinary practice, including  dexmedetomidine (dexdomitor).
  2. To explore the possibilities of pharmacological provocation of epileptic activity…

12) Can the authors explain why there were no controls for each of the groups (epileptic and non-epileptic) with saline injections for example, instead of drugs? 

Response: This is one of the limitations of our study. We have discussed these issues in Section 3.4 “Limitations and further directions”.

13) The authors should specify the exact number of experimental units allocated to each group, and the total number in each experiment. Also indicate the total number of animals used. 

Response: According to the standard template, methods should be placed after results, and this can have a negative impact on readability. The total number of rats (n=26) is indicated in Results on page 5 and in Methods on page 18.
The exact number of experimental units in WAG/Rij arts is indicated in Figure 5 b (right table).  The exact number of experimental units in NEW rats can be found ‌in Supplementary S2 (Table S2.1, Figure S2.1).

14) Were there any animals lost during the experiments? 

Response: No animals were harmed or lost during experiments, because Dex was used in therapeutic doses. This was added to Methods (Section 4.1, page 20) In two animals, EEG sets on their heads were broken after one baseline record followed by a pharmaco-test. 

15) Please provide how was sample size calculated. 

Response: In the current study we investigated the effect of different i.p. doses of dexdomitor on epileptic spike-wave activity in WAG/Rij rats and in a new non-epileptic substrain (NEW). This kind of study has not been done before. Inasmuch as no previously obtained data were available, it was not possible to assume the effect of sample size and the standard deviation. IIn this case, the “resource equation” method can be used to calculate the E value. After the clusterization of the SWDs scores in the epileptic group, we finally defined 3 clusters for 13 rats. Thus, E=13-3=10. The corresponding E-value for non-epileptic rats was E=13-1=12. The E-value between 10 and 20 indicates that the sample was not too big and not too small.

16) Was the power of the study calculated? 

Response: We did not calculate the power of the study, because it was an explorative and descriptive study. We had two proposals: (1) Dex injections would not elicit SWDs in non-epileptic subjects; (2) a dosage of Dex could be found that would lead to a prolonged increase in SWDs, without initiating sedation. In both cases, an effect that can be present or not, regardless of the number of positive and negative results. Therefore, our study was purely descriptive. With this approach we discovered two cases of latent epilepsy and several cases of prolonged SWD spike-wave activity provocation in epileptics (the Type 2 response).  

 17) Was data distribution normal? 

Response: The Kolmogorov-Smirnov test was used for each sample to test for normality of the distribution. In the case of n<10 we automatically decided for non-parametric statistical tests in the further analysis. This information is now described in Section 4.5.

18) The authors should provide exclusion and inclusion criteria for the animals during the experiments data points during the analysis. Were these criteria stablished previously? 

Response: For the epileptic group, the inclusion criterion was the presence of spontaneous genuine SWDs (three and more SWDs per hour).
We clearly stated this in the first paragraph of Introduction (page 5): “The inclusion criterion for the epileptic group: higher than 3 genuine SWDs per hour in a full length baseline record. There were no exclusion criteria for the epileptic group.” See details in Table S3.1 in Supplementary S3. 
“The inclusion criterion for the non-epileptic group: the number of genuine SWDs in baseline was less than 0.25 per hour (1 SWD per 4 hours). The exclusion criterion: the number of genuine SWDs in baseline was higher than 3 per hour”.

19) In each experiment, authors should report any animals or data points not included in the analysis and explain why. 

Response: We statistically analyzed all data points. Our rats showed different responses to Dex, and we classified them (K-means clustering) in order to identify types of reactions.  All data points are shown in ‌revised Fig. 6. Individual data for cluster analysis were presented in Table S3.2, Supplementary S3. Supplementary S3 also displays the individual results of the SWDs density in the full-length time, which are summarized in Fig. 6.

20) I advise the authors to report the exact number of animals in each experimental group. 

Response: The exact number of rats in each experimental group has been reported in each study part.  

21) The authors should state how they reduced the potential biases in the experiments such as environmental factors, the order of treatments and measurements, or animal/cage location. 

Response: Animals were housed in the institute’s vivarium with the 12/12 h light-dark cycle and light onset time set on 8 a.m. All rats were born, raised, and given all the same conditions in the same environment. After the surgery, the rats were allowed to recover for 10 days, and they were kept in individual cages on the same two lower rows. Experiments were performed in the sound attenuated room with the same characteristics as in the vivarium. We have also performed baseline recording before each Dex injection, and it helped to reduce a part of potential biases. We controlled the estrus cycle phase in female rats. This information has now been added to Section 4.1.

22) 23) Was this study blinded? Were the researchers aware of group allocation during the experiments? 

Response: The researchers were aware of experimental details, and therefore the study was not blinded. Dex was injected in random doses, and SWDs were defined automatically. It was not necessary to be blinded to ‌information about experimental procedures, because SWDs were detected automatically in the full length raw EEG. We cover this on page 5: “The threshold values, which were selected to detect SWDs in baseline, were also used to detect SWDs in Dex conditions. This provided accurate and reliable detections, which helped to overcome the significant drawback of subjectivity.” 

24) All of the outcome measures should be clearly defined. 

Response: We sincerely apologize for the inaccurate and inappropriate presentation of outcomes. We carefully revised the Results and Materials and Methods sections and added missed information. 

25) What was the primary outcome measure? Was it used to calculate sample size? 

Response: The number of SWDs , SWDs density and the mean duration of SWDs were computed. For each rat we computed the difference in dSWD values between baseline and Dex conditions in 5 min bins. For the vigilance states analysis we used the latency of the first SWD after the injection (L1swd), the latency and the total duration of sedative state (L2 and D2 respectively), the duration of the deep sedation phase (if present, D3), the duration of the 1st and the 2nd period of enhanced spike-wave epileptic activity (D1swd and D2swd respectively). We did not use these measures to calculate sample size (see also question 15).

26) The statistical analysis section needs significant improvement and further detailing. Which statistical packages were used? 

Response: We used Statistica 12. This is now added to Methods, section 4.5. “Statistical Analysis”. The section has been completely revised and updated to reflect full details of data analysis.

27) Provide more details about each statistical analysis done, including confidence intervals, effect size and appropriate parameters. I suggest seeking advice from a professional statistician for statistics reporting in this paper since this section is extremely important and needs significant improvement. 

Response: We agree with this critical remark. Descriptions of some statistical tests were missing in Section 4.5 “Statistical Analysis”.  We added comprehensive information about the tests used in our study. The data about necessary parameters was added to Results. We have not evaluated the effect size in any case.

28) Please provide more details about how were seizures analyzed. Was any scale used? 

Response:  We apologize for the limited methodological description. Our manuscript appears too long, and we had to describe methodological details in four Supplementaries. In the revised MS at the beginning of Results we added a brief description:  “SWDs were detected when wavelet power at 8-10 Hz and 17-20 Hz exceeded individually chosen threshold values (Supplementary S1). The threshold values, which were selected to detect SWDs in baseline, were also used to detect SWDs in Dex conditions (Section 4.3.2). This provided accurate and reliable detections, which helped to overcome the significant drawback of subjectivity. We analyzed datasets containing the start and end points of SWDs and computed the following scores” More details are given in Section 4.3.2 and in Supplementary S1. 

29) Can the authors provide video-recordings of the experiments? 

Response: An example of video-recordings is shown in video-entry at https://encyclopedia.pub/video/video_detail/628

30) Was video-EEG performed? 

Response: Yes, video-EEG was recorded in each rat used for the assessment of EEG/behavioral states. This issue has now been addressed in Section 4.3.1.

31) Please describe any methods used to assess whether the data met the assumptions of the statistical approach, and what was done if the assumptions were not met. 

Response: In order to choose the statistical approach, at the first step we tested the assumption that the variable distribution was normal. The Kolmogorov-Smirnov test was used to test for normality of the distribution (p>0.05 for a normal distribution). Normally distributed data were analyzed with a repeated measures ANOVA with Duncan post-hoc test. Non-normally distributed data were analyzed with non-parametric tests such as the Friedman ANOVA, Mann Whitney U test, Wilcoxon matched pairs test and Pearson correlation test. This has been added to Section 4.5. (Statistical Analysis).

32) More details should be provided about the experimental animals such as age or developmental stage. 

Response: This information has now been added to the MS. We added used rats that were older than 6 months when WAG/Rij rats were known to develop spontaneous SWDs [van Luijtelaar & Coenen, 1986; Sitnikova et al, 2015; Lazarini-Lopes et al., 2021 Coenen 1987; van Luijtelaar & Oijn, 2020; Lazarini-Lopes 2021]. This has been added to the Introduction.

33) Please provide further relevant information on the provenance of animals, health/immune status, genetic modification status, genotype, and any previous procedures. 

Response: This information has also been added to the MS. Section 2, page 4:  “All of the rats were bred and raised at our Institute, and they were all intact and not genetically modified.” 

34) The authors should further specify timing and frequency of procedures for reproduction purposes. 

Response: All rats were bred and maintained at the Institute of Higher Nervous Activity and Neurophysiology of RAS, Moscow (IHNA). In our experiments, we controlled ‌the estral cycle in females and defined diestus, proestrus, estrus and metestrus. The female rats did not give birth before or during the current experiment.

35) Please include where procedures were carried out (including detail of any acclimatization periods). 

Response: All procedures (including recording during baseline and after administration of Dex) were carried out in the experimental room under environmentally controlled conditions with a 12:12 h light:dark cycle (lights on at 8 a.m.), with constant ventilation and airing. Rats were provided with unlimited access to food and water. The environmental conditions were the same as those in the vivarium. The rats were not additionally acclimatized to the experimental room. This information has been added to Section 4.1.

36) For each experiment conducted, including independent replications, report summary/descriptive statistics for each experimental group, with a measure of variability where applicable (e.g., mean and SD, or median and range). 

Response:  In the revised version, we have reported mean and SD for normally distributed data, or median and range for non-normally distributed data.

37) The authors should describe any interventions or steps taken in the experimental protocols to reduce pain, suffering, and distress. 

Response: The procedures that were intended to reduce pain, suffering, and distress are indicated in the original protocol in Russian (uploaded to Google-doc) https://docs.google.com/document/d/14zDIje5xmhBPueAxo7QszLztJRSYjteK/edit?usp=sharing&ouid=100065682202293471689&rtpof=true&sd=true

  1. Treatment of the area of surgical intervention with streptocide powder. Intramuscular injection of metamizole for pain relief after surgery (FSSCI Microgen, Russia, dose 25 mg/kg).
  2. Prior to recovery from anesthesia, the animal will be kept on a heated surface (37 degrees Celsius) to prevent a drop in body temperature prior to recovery from anesthesia.
  3. After recovery from anesthesia, the animal will be moved to a clean home cage with free access to water and food.

  1. The recovery period = 7-10 days after the operation, during which daily monitoring of the animal's condition will be carried out, treatment of the area of surgical intervention with a 3% hydrogen peroxide solution.

In the revised MS we briefly stated “Care was taken to reduce pain, suffering, and distress as indicated in the experimental protocol approved by the animal ethics committee of IHNA RAS.” 

38) In the discussion, authors should add the study limitations, including potential sources of bias, limitations of the animal model, and imprecision associated with the results.

Response: We added Section 3.4, "Limitations and further directions", where we discussed the weak points of our research.

Reviewer 3 Report

This article presents data clearly and it represents an original contribution. The abstract and title are informative and reflect the content of the paper, appropriate keywords have been given.

The introduction is extensive, contains well-selected and cited literature, and is sufficient to explain the topics of the paper. Perhaps even too extensive in relation to the discussion.

Methods and results are sound and relevant. Figures nicely present the results.

However, I would recommend some corrections:

Something is missing in the first sentence on page 15. 

Weaknesses and limitations of the data should be pointed out and discussed. 

It is not sufficiently explained why which mouse received which dose. 

The English language is fine, but it is necessary to correct a few typos and correct certain sentences.

Author Response

Dear Editor, 
We are extremely grateful for the time and effort that you dedicated to reviewing our paper. Thank you for your positive feedback and suggestions.

Something is missing in the first sentence on page 15.
Response: We apologize for submitting the first version of our manuscript to the journal without going through a rigorous proofreading process. In the revised version, we made a number of improvements to the text's readability and grammatical accuracy, including fixing spelling and grammar errors.

Weaknesses and limitations of the data should be pointed out and discussed.
Response: We added Section 3.4, "Limitations and further directions", where we discussed the weak points and limitations of our research.

It is not sufficiently explained why which mouse received which dose.
Response: The drug (Dexmedetomidine) was randomly injected in doses ranging from 0.004 to 0.034 mg/kg to cover all possible outcomes. This issue has been addressed in Section 4.2: "Dexmedetomidine was injected i.p. in different therapeutic doses ranging from 0.003 to 0.049 mg/kg. In order to select the most effective dose with the highest pro-epileptic effect, i.e., the central dose, we did pilot experiments in epileptic rats. The effect of Dex was individual, and the doses that were given varied across the central dose. Non-epileptic rats were injected with the same (central) dose of Dex." 

Round 2

Reviewer 2 Report

The authors have replied sufficiently to the reviewer's comments.

Minor editing required.